# Mitochondrial Respiratory Defect Enhances Hepatoma Cell Invasiveness via STAT3/NFE2L1/STX12 Axis

**DOI:** 10.3390/cancers12092632

**Published:** 2020-09-15

**Authors:** Young-Kyoung Lee, So Mee Kwon, Eun-beom Lee, Gyeong-Hyeon Kim, Seongki Min, Sun-Mi Hong, Hee-Jung Wang, Dong Min Lee, Kyeong Sook Choi, Tae Jun Park, Gyesoon Yoon

**Affiliations:** 1Department of Biochemistry, Ajou University School of Medicine, Suwon 16499, Korea; algamsa109@ajou.ac.kr (Y.-K.L.); smkwon@ajou.ac.kr (S.M.K.); leb416@ajou.ac.kr (E.-b.L.); zxcv763@ajou.ac.kr (G.-H.K.); tjdrl4100@ajou.ac.kr (S.M.); hsmfreesia@ajou.ac.kr (S.-M.H.); ldonglminl@ajou.ac.kr (D.M.L.); kschoi@aumc.ac.kr (K.S.C.); park64@ajou.ac.kr (T.J.P.); 2Department of Biomedical Sciences, Graduate School, Ajou University, Suwon 16499, Korea; 3Department of Surgery, Ajou University School of Medicine, Suwon 16499, Korea; wanghj@ajou.ac.kr

**Keywords:** hepatocellular carcinoma, mitochondrial defect, NFE2L1, retrograde signaling, STAT3, STX12

## Abstract

**Simple Summary:**

Mitochondria are essential organelles responsible for aerobic ATP production in eukaryotes. However, many solid tumor cells harbor an impaired mitochondrial ATP production system: oxidative phosphorylation (OXPHOS). The aim of this study was to elucidate the involvement of the mitochondrial OXPHOS defect in cancer cell activity, especially focusing on hepatoma cell invasiveness. We demonstrated that NADH:Ubiquinone Oxidoreductase Subunit A9 (NDUFA9) depletion was an upstream driver of the OXPHOS defect and nuclear factor-erythroid 2 like 1 (NFE2L1) upregulation in HCC tumors. NFE2L1 is the key transcription factor to enhance hepatoma cell invasiveness via STX12 expression. Our study presents a novel mitochondrial dysfunction-mediated retrograde signaling pathway and the resulting transcriptomic reprogramming in liver cancer progression, providing the NDUFA9/NFE2L1/STX12 axis as a key prognostic marker of aggressive liver cancer with mitochondrial defects.

**Abstract:**

Mitochondrial respiratory defects have been implicated in cancer progression and metastasis, but how they control tumor cell aggressiveness remains unclear. Here, we demonstrate that a mitochondrial respiratory defect induces nuclear factor-erythroid 2 like 1 (NFE2L1) expression at the transcriptional level via reactive oxygen species (ROS)-mediated STAT3 activation. We identified syntaxin 12 (STX12) as an effective downstream target of NFE2L1 by performing cDNA microarray analysis after the overexpression and depletion of NFE2L1 in hepatoma cells. Bioinformatics analysis of The Cancer Genome Atlas Liver Hepatocellular carcinoma (TCGA-LIHC) open database (*n* = 371) also revealed a significant positive association (*r* = 0.3, *p* = 2.49 × 10^−9^) between NFE2L1 and STX12 expression. We further demonstrated that STX12 is upregulated through the ROS/STAT3/NFE2L1 axis and is a key downstream effector of NFE2L1 in modulating hepatoma cell invasiveness. In addition, gene enrichment analysis of TCGA-LIHC also showed that epithelial–mesenchymal transition (EMT)-related core genes are significantly upregulated in tumors co-expressing NFE2L1 and STX12. The positive association between NFE2L1 and STX12 expression was validated by immunohistochemistry of the hepatocellular carcinoma tissue array. Finally, higher EMT gene enrichment and worse overall survival (*p* = 0.043) were observed in the NFE2L1 and STX12 co-expression group with mitochondrial defect, as indicated by low NDUFA9 expression. Collectively, our results indicate that NFE2L1 is a key mitochondrial retrograde signaling-mediated primary gene product enhancing hepatoma cell invasiveness via STX12 expression and promoting liver cancer progression.

## 1. Introduction

Deregulated cellular energetics is one of the hallmarks of cancer [1]. The bioenergetic deregulation is mainly characterized by defective or decreased mitochondrial oxidative phosphorylation (OXPHOS) and a gain of energy dependence on glycolysis, even in the presence of oxygen. This is also termed ‘aerobic glycolysis’ and was initially proposed by Otto Warburg [2,3]. Upregulated glycolysis and resulting lactate production strongly support diverse cancer cell activities, including cell proliferation, angiogenesis, tumor invasion activity, and chemoresistance, thereby promoting tumor malignancy [4,5]. However, the causal link of mitochondrial OXPHOS dysfunction with tumor formation and development has not been clearly understood, although accumulating evidence that mitochondrial dysfunction is associated with and promotes cancer progression has been reported recently [3,6,7,8]. Deregulated mitochondria intimately communicate with the nuclear gene expression system by transmitting diverse retrograde signals to restore their function or modify the cellular fate to death, senescence, or proliferation [9,10,11,12,13]. Therefore, to unveil the mechanistic link between mitochondrial dysfunction and cancer progression, it is critical to elucidate the mitochondrial retrograde signaling-mediated transcriptional regulation in detail as well as its contribution to cancer cell activity.

Nuclear factor-erythroid 2 like 1 (NFE2L1, also known as Nrf1) is a transcription factor (TF) belonging to the cap and collar-basic leucine zipper (CNC-bZIP) family, which also includes Nrf2, a canonical TF modulating the antioxidant response [14,15]. NFE2L1 is ubiquitously expressed in a broad range of tissues and acts as a key regulator of diverse cellular responses, including the antioxidant response, lipid metabolism, and proteasome homeostasis [16,17,18]. Global NFE2L1 -knockout mice are embryonic lethal at the mid-gestation stage [19], and neuron-specific NFE2L1 loss occurs due to proteasomal deregulation and neurodegeneration [20]. However, hepatocyte-specific NFE2L1-knockout mice exhibit severe hepatic steatosis and liver carcinoma [21]. These findings suggest that NFE2L1 differentially modulates diverse tissue-specific homeostasis and plays a role in the development of liver cancer. In an effort to elucidate the mechanistic role of mitochondrial dysfunction in liver cancer progression, we recently reported commonly regulated transcripts in response to diverse mitochondrial defects using several hepatoma cell lines. NFE2L1 is one of the commonly upregulated TFs and a potential regulator of hepatoma cell invasiveness [22].

In this study, we demonstrate that the mitochondrial OXPHOS defect enhances NFE2L1 transcription via reactive oxygen species (ROS)-mediated STAT3 activation, and the upregulation of NFE2L1 increased hepatoma cell invasiveness by inducing syntaxin 12 (STX12) expression. We validated our results by performing analysis using the transcriptome datasets of The Liver Hepatocellular Carcinoma project of The Cancer Genome Atlas (TCGA-LIHC). Collectively, our results indicate that NFE2L1 is a key mitochondrial retrograde signaling-mediated gene product enhancing hepatoma cell invasiveness via STX12 expression.

## 2. Results

### 2.1. NFE2L1 is Upregulated by OXPHOS Defect and Regulates Hepatoma Cell Invasiveness

Previously, we reported that NFE2L1 mRNA expression is commonly upregulated in response to diverse mitochondrial defects, such as OXPHOS inhibition and mitochondrial DNA depletion [22]. To elucidate the functional and mechanistic involvement of NFE2L1 in hepatoma cell activity, we examined protein levels in several hepatoma cell lines with (SNU354 and SNU423) or without (SNU387 and Ch-L) mitochondrial defects. Compared to SNU387 and Ch-L cells, SNU354 and SNU423 hepatoma cells had higher NFE2L1 protein levels (Figure 1A), corresponding to the NFE2L1 mRNA level (Figure 1B). We also observed that SNU354 and SNU423 have lower levels of NDUFA9 protein, which is a subunit of OXPHOS complex I (Figure 1A), and lower oxygen consumption rate (OCR) activities (both ATP-linked basal OCR and maximal OCR including spare capacity) than SNU387 and Ch-L cells (Figure 1C). However, SNU354 and SNU423 possessed increased extracellular acidification rates, extracellular lactate levels, and glucose consumption rates (Figure 1D,E), eventually showing increased cellular ATP levels (Figure 1F). NDUFA9 knockdown was enough to suppress complex I-associated OCR (Figure 1G, right) [23] and increased NFE2L1 mRNA expression (Figure 1G, left). These results imply that NDUFA9 depletion may be one of the causes of OXPHOS defect and an upstream event of NFE2L1 expression in hepatoma cells. An inverse association between NFE2L1 and NDUFA9 expression in HCC was found by analyzing the RNA-seq data of TCGA-LIHC (*n* = 371; Figure 1H). When we monitored NFE2L1 and NDUFA9 protein levels using six paired HCC and adjacent tissue samples, four cases showed clearly increased NFE2L1 expression in tumor compared to the surrounding tissue. Three of these cases had decreased NDUFA9 expression (Figure 1I), supporting the results of the TCGA analysis. Next, we compared the NFE2L1 expression levels among tissue types with clinical aggravation in TCGA-LIHC and observed a tendency of increasing NFE2L1 from normal liver tissue to primary tumor and recurrent tumor (Figure 1J). Thus, we asked whether the NDUFA9-associated NFE2L1 expression contributed to hepatoma cell invasiveness. We divided the primary LIHC samples into two groups based on the median expression of both NFE2L1 and NDUFA9: high NFE2L1 and low NDUFA9 expression (NFE2L1-high/NDUFA9-low) and low NFE2L1 and high NDUFA9 expression (NFE2L1-low/NDUFA9-high). Then, we compared the enrichment of the epithelial–mesenchymal transition (EMT) signature genes between the groups. The NFE2L1-high/NDUFA9-low group was more enriched in EMT signature genes than the other group (Figure 1K), implying that the NDUFA9-associated NFE2L1 expression may play a key role in hepatoma cell invasiveness. Furthermore, siRNA-mediated NFE2L1-knockdown in SNU423 cells significantly diminished the cell invasion activity without affecting cell growth and NDUFA9 expression (Figure 1L–N). Taken together, these results indicate that NFE2L1 is a key regulator of hepatoma cell invasiveness, and NDUFA9 depletion is an upstream driver of OXPHOS defect and NFE2L1 upregulation in HCC tumors.

### 2.2. Mitochondrial ROS Increases NFE2L1 Expression via Transcriptional Activation

Next, we addressed how the OXPHOS defect regulates NFE2L1 gene transcription. SNU354 and SNU423 cells, which harbored low OCR activity, expressed higher levels of both mitochondrial and total cellular ROS than SNU387 and Ch-L cells (Figure 2A,B). When the intracellular ROS level was increased by exposing Ch-L cells to exogenous H_2_O_2_, both the mRNA and protein levels of NFE2L1 were significantly increased (Figure 2C,D). NFE2L1 acts as a TF after being processed in the endoplasmic reticulum (ER) and translocated into the nucleus [15]. Therefore, we examined alterations in the cellular localization of NFE2L1 in response to intracellular ROS. In Ch-L cells, NFE2L1 was basally dispersed in both the nucleus and cytoplasm, and nuclear NFE2L1 was further increased by H_2_O_2_ treatment (Figure 2E). The cytoplasmic and nuclear localization of NFE2L1 was further confirmed by its overexpression (Figure 2E, bottom). Notably, some cells were enlarged by 200 μM H_2_O_2_ treatment. The time course response of NFE2L1 mRNA expression to H_2_O_2_ treatment was observed in both Ch-L and SNU387 cells (Figure 2F,G). SNU387 cells were more vulnerable to H_2_O_2_ than Ch-L cells. Therefore, we used 50 μM H_2_O_2_ for SNU387 in Figure 2G. In addition, when we used N-acetyl cysteine (NAC) to decrease the intracellular ROS level in SNU423 cells, which harbor high ROS levels, NFE2L1 expression was clearly decreased (Figure 2H,I).

To monitor NFE2L1 promoter activity, we generated a luciferase reporter plasmid harboring 1899 bp of the NFE2L1 promoter sequence (Figure 2J). We previously reported that the pharmacological inhibition of OXPHOS complex II by 2-thenoyltrifluoroacetone (TTFA) induces continuous mitochondrial ROS generation and increases intracellular ROS levels [24]. When we increased the intracellular ROS using TTFA in Ch-L cells, the NFE2L1 promoter activity was significantly augmented (Figure 2K). Reducing the intracellular ROS level by NAC treatment suppressed the promoter activity, whereas augmenting the ROS level by H_2_O_2_ increased the activity in SNU423 cells (Figure 2L,M). These results indicate that an OXPHOS defect enhanced NFE2L1 promoter activity and expression via ROS generation.

### 2.3. ROS-Mediated STAT3 Activation is Key Mitochondrial Retrograde Signaling for NFE2L1 Expression

To elucidate the mechanism of how ROS regulates NFE2L1 expression, we searched potential TF binding sites in the NFE2L1 promoter sequences between −1580 and +318 using the LASAGNA-Search program [25]. Among the potential TFs, we selected 11 TFs that have more than two possible binding sites in the promoter. When we individually depleted the selected TFs in SNU423 cells, which have high NFE2L1 levels, the NFE2L1 mRNA level was decreased only by knocking down four TFs, such as USF2, STAT3, JUN, and SREBP1 (Figure 3A). Next, we monitored the endogenous protein levels of the four TFs in the hepatoma cell lines. SNU354 and SNU423 cells harbored low levels of SREBP1 and USF2, high levels of JUN expression, and minimally increased STAT3 protein levels with highly phosphorylated status (Figure 3B). When the four TFs were individually depleted in SNU423, we observed that only STAT3 depletion decreased NFE2L1 protein expression (Figure 3C). STAT3 activation by the overexpression of wild-type STAT3 increased NFE2L1 mRNA and protein levels in both Ch-L and SNU387 cells (Figure 3D–F). Blocking STAT3 activation using pharmacological inhibitor AG490 [26] or by introducing siRNA diminished NFE2L1 expression in SNU423 cells (Figure 3G,H). In addition, siRNA-mediated STAT3 knockdown and the overexpression of dominant-negative (dn) STAT3 mutant (Y705F) significantly diminished NFE2L1 promoter activity (Figure 3J,K). The overexpression of wild-type STAT3 augmented the activity of both the 1899-bp and 443-bp NFE2L1 promoter in Ch-L cells (Figure 3L). These results demonstrate that STAT3 activation is a key upstream regulator of NFE2L1 transcription. Next, we examined whether ROS-induced NFE2L1 expression is mediated through STAT3 activation. The exposure of Ch-L cells to H_2_O_2_ increased STAT3 phosphorylation (activation) and NFE2L1 expression in a dose-dependent manner (Figure 3M), and the H_2_O_2_-induced STAT3 activation and NFE2L1 induction was further enhanced by wild-type STAT3 overexpression but suppressed by dn-STAT3 (Y705F) overexpression (Figure 3N). These results indicate that ROS-mediated STAT3 activation enhances NFE2L1 expression.

### 2.4. STX12 is a Key Downstream Effector Gene of NFE2L1

To identify the downstream effector gene of NFE2L1 in the modulation of hepatoma cell invasiveness, transcriptome profiling was performed using a cDNA microarray after NFE2L1 was suppressed in SNU423 cells and overexpressed in SNU387 cells. We selected genes downregulated by NFE2L1 knockdown or/and upregulated by NFE2L1 overexpression (fold difference > 0.3). The selected genes were compared to the commonly upregulated genes in hepatoma cells with mitochondrial defects (SNU354 and SNU423) compared to hepatoma cells without mitochondrial defects (Ch-L and SNU387), which were previously reported [22]. From this analysis, 10 commonly regulated genes were identified (Figure 4A,B), and we validated the role of four genes (ENO2, EphB2, SerpinE1, and STX12), which were reportedly involved in cancer cell activities [27,28,29,30], as downstream targets of NFE2L1. NFE2L1 knockdown in SNU423 cells decreased ENO2, SerpinE1, and STX12 expression at both the mRNA and protein levels (Figure 4C,D). NFE2L1 overexpression in SNU387 cells increased all four genes at the mRNA level, but they increased EphB2, SerpinE1, and STX12 at the protein level (Figure 4E,F). Interestingly, among the 10 commonly regulated genes, only STX12 exhibited a significant positive association with NFE2L1 in TCGA-LICC cohort (*n* = 371; Figure 4G). Moreover, when we compared STX12 expression between the two NFE2L1/NDUFA9 groups in Figure 1G, we found that STX12 expression was significantly upregulated in the NFE2L1-high/NDUFA9-low group (Figure 4H). These results suggest that STX12 may be the key downstream effector of NFE2L1 in NDUFA9-mediated OXPHOS-defective HCC.

### 2.5. STX12 Expression is Regulated by the ROS/STAT3/NFE2L1 Axis

Next, we examined whether STX12 is truly regulated by ROS-mediated STAT3 activation and its downstream NFE2L1 expression. SNU354 and SNU423 cells possessed high levels of STX12 protein (Figure 5A). The exposure of Ch-L cells to H_2_O_2_ increased STX12 mRNA levels in both a dose- and time-dependent manner (Figure 5B,C). The dose-dependent response of STX12 protein was also confirmed, similar to the patterns of STAT3 phosphorylation and NFE2L1 expression (Figure 5D). STX12 induction by H_2_O_2_ was also observed in SNU387 cells (Figure 5E). In addition, STAT3 overexpression increased STX12 at the protein and mRNA levels in Ch-L cells, and STAT3 depletion decreased STX12 mRNA in SNU423 cells (Figure 5F–H). Furthermore, the STX12 increase induced by H_2_O_2_ was reversed by the STAT3 inhibitor AG590 (Figure 5I), supporting the involvement of STAT3 in ROS-mediated STX12 expression. The overexpression of NFE2L1, the downstream TF of STAT3, increased STX12 from the mRNA level in Ch-L cells, and its depletion suppressed STX12 mRNA in SNU423 cells (Figure 5J,K). STX12 induced by H_2_O_2_ was also suppressed by NFE2L1 knockdown in Ch-L cells (Figure 5L). These results support our hypothesis that STX12 expression is regulated by the ROS/STAT3/NFE2L1 axis.

### 2.6. STX12 is a Key Regulator of Hepatoma Cell Invasiveness

Next, we investigated the role of STX12 in hepatoma cell invasiveness. When STX12 was depleted in SNU423 cells, the cell invasion activity was suppressed without affecting the cell growth rate (Figure 6A,B), implying its involvement in cell invasiveness. STX12 overexpression induced the cell invasion activity of Ch-L cells (Figure 6C,D). To evaluate the long-term effect, we generated NFE2L1-depleted SNU423 cells using the CRISPR-Cas9 system. The NFE2L1-depleted cells exhibited a further effective decrease in cell invasion activity (Figure 6E–G). However, overexpression of STX12 in the cells restored invasion activity, but not strongly (Figure 6E–G). These results imply that STX12 is a downstream effector of NFE2L1 in the modulation of hepatoma cell invasiveness.

Next, to validate the relationship between NFE2L1 and STX12 protein expression, we performed immunohistochemistry using a liver cancer tissue array. A higher expression (level 2) of STX12 was observed in the tissues with high NFE2L1 expression (Figure 6H). We analyzed again the clinical importance of these findings using the TCGA-LIHC cohort database. When comparing enrichment of the EMT signature based on the expression of NFE2L1 or STX12, the high-expression group of NFE2L1 and/or STX12 exhibited a more enriched EMT signature than the low-expression groups (Figure 6I). To more precisely interrogate the effect of OXPHOS defect on the NFE2L1/STX12-linked EMT profile, we subdivided the groups elected by NFE2L1 and STX12 levels into group H and group L based on the NDUFA9 level. In detail, group H had high NFE2L2 and STX12 but low NUDFA9, implying a high OXPHOS defect-linked feature, and vice versa for group L. Interestingly, group H had a higher enrichment of the EMT signature (Figure 6J). Furthermore, when we compared clinical outcomes between the two groups by using Kaplan–Meier survival analysis, we observed that group H had significantly worse overall survival (Figure 6K). Taken together, these results indicate that NFE2L1/STX12 expression is the key to regulating hepatoma cell invasiveness and may be useful as a prognostic marker for HCC.

## 3. Discussion

Bioenergetic rewiring to impaired mitochondrial OXPHOS activity with augmented glycolytic flux is a key feature of many cancers, including HCC. The impaired OXPHOS activity modifies mitochondria-to-nuclear communication, which is called retrograde signaling, and it subsequently reprograms nuclear gene expression, thereby modifying cellular function and fate [9,10,13]. The reprogrammed gene products provide cells with a new driving force to acquire various cancer cell activities, such as neoplastic transformation, aggressiveness, and acquired drug resistance [22,31,32,33]. The present study demonstrates that NFE2L1 is one of the reprogrammed gene products critically involved in hepatoma cell invasiveness via the expression of STX12. Multifaceted roles of NFE2L1 in cellular homeostasis have been reviewed recently [15], but the current understanding of its pathological involvement in cancer development is limited. NFE2L1 also exhibits tissue-dependent expression despite its ubiquitous presence in diverse tissue types. Brain and skin express high levels of NFE2L1, whereas small intestine and stomach express low levels [15]. The liver has a moderate level of NFE2L1 mRNA according to GeneAtlas. Interestingly, liver-specific NFE2L1 inactivation results in steatohepatitis, fibrosis, and the spontaneous development of hepatic neoplasia in mice [21], implying its potential negative involvement in liver cancer development. In contrast, NFE2L1 transgenic mice present with impaired glucose metabolism in the liver and insulin resistance in a diet-induced obesity model [34]. However, our results clearly demonstrate that augmented NFE2L1 expression grants invasion activity to immortalized liver cells (Ch-L) and non-aggressive hepatoma cells (SNU387), suggesting a differential role of NFE2L1 according to cellular context.

We found that NFE2L1 upregulation is achieved through a retrograde signaling pathway comprising mitochondrial ROS-mediated STAT3 activation. This finding emphasizes the involvement of mitochondrial OXPHOS defect and consecutive retrograde signaling-mediated transcriptomic reprogramming in malignant HCC. To the best of our knowledge, our present study is the first to demonstrate the mechanistic link between NFE2L1 transcription and mitochondrial retrograde signaling, especially in HCC.

Although NFE2L1, which is synthesized de novo in response to mitochondrial dysfunction, has been shown to control hepatoma cell invasion activity, NFE2L1 itself is a TF. This suggests that NFE2L1 cannot directly regulate the invasion activity and manifests its controlling power via the synthesis of secondary effector molecule(s) responsible for the hepatoma cell invasiveness. Identification of the newly expressed secondary effectors is crucial to elucidating the underlying mechanism of mitochondrial dysfunction-driven hepatoma cell invasiveness. Reportedly, NFE2L1 regulates the GSH-mediated antioxidant response by expressing Nqo1, Gclc, and Gclm [35,36], proteasome activity by regulating Herpud1 and PsmB6 [17,18,37,38], and lipid metabolism through Lipin1 and PGC-1b expression [17]. In this study, we presented STX12 as a key downstream driver of NFE2L1 to induce hepatoma cell invasiveness. Although limited information is available on the biological function of STX12 (also known as STX13), it was first identified as being localized in the endosome [39] and involved in the SNARE-dependent trafficking of matrix metalloproteinases, thereby remodeling the extracellular matrix and modulating the cell invasion activity of human fibrosarcoma cells [28,40]. These findings support the role of STX12 as a key downstream effector target of NFE2L1 to modulate hepatoma cell invasiveness. The knockdown of STX12 in SNU423 cells suppressed cell invasiveness, whereas the overexpression of STX12 in Ch-L cells induced invasiveness, implying that STX12 is an essential and sufficient regulator of hepatoma cell invasiveness. Notably, the modulation of STX12 expression did not affect the cell growth rate, indicating its specific involvement in the invasion activity. The positive association of EMT gene enrichment with the high expression of both NFE2L1 and STX12 in the bioinformatics analysis of the TCGA-LIHC database further supports the involvement of STX12 in HCC progression. Collectively, our results indicate the potential for using NFE2L1/STX12 as an effective prognostic marker of HCC malignancy with mitochondrial defect at diagnosis.

## 4. Materials and Methods

### 4.1. Cell Culture, Cell Growth Rate, and Tumor Samples

The human hepatoma cell lines (SNU354, SNU387 and SNU423) were purchased from the Korean Cell Line Bank (KCLB, Seoul, Korea) and cultured in Gibco^TM^ RPMI 1640 Medium (Cat. 31800022, Invitrogen, Thermo Fisher Scientific Inc., Waltham, MA, USA), supplemented with 10% fetal bovine serum (Invitrogen) and Gibco^TM^ Antibiotic-Antimycotic (Cat. 15240062, Invitrogen) at 37℃, in a humidified incubator with 5% CO_2_. To obtain Ch-L clones that have higher liver cell characteristics, single cell clones were isolated using a limiting dilution and expansion of Chang cells (American Tissue Culture Collections, Rockville, MD, USA) as previously reported [22]. For this study, the liver features were again validated by albumin and carbamoyl phosphate synthetase 1 (CPS1) expression (Appendix A). A Ch-L clone that has higher liver cell characteristics was selected and cultured in Dulbecco’s modified Eagle’s medium (DMEM, Invitrogen) supplemented with 10% fetal bovine serum and antibiotics.

Cell growth rates were monitored by counting the cells after staining with 0.4% (w/v) trypan blue (Invitrogen). At the end of each experiment, viable cells (trypan blue-negative cells) were counted, using the Countess™ automated cell counter (Invitrogen) to exclude dead cells. Overall, dead cells (trypan blue positive cells) were not significantly found in this study.

Paired tumor and surrounding non-tumor tissue samples (freshly frozen specimens) were obtained from the Ajou Human Bio-Resource Bank (AHBB), which is a member of the National Biobank of Korea that is supported by the Ministry of Health and Welfare. No patient received chemotherapy or radiation therapy before the surgery.

### 4.2. Monitoring Cellular Oxygen Consumption Rate (OCR) and Extracellular Acidification Rate (ECAR)

Cellular OCR and ECAR were monitored using a Seahorse XF24 Extracellular Flux Analyzer (Seahorse Bioscience, Santa Clara, CA, USA) according to the provided protocol, with slight modification. Briefly, cells (1 × 10^4^) were seeded on XF24 cell culture microplates (Seahorse Bioscience) and preincubated overnight (37 °C, CO_2_-free) with XF base medium (Seahorse Bioscience) containing 1 mM sodium pyruvate (Invitrogen), 4 mM glutamax (Invitrogen), and 11 mM glucose (Invitrogen) to culture medium condition. Basal OCR (ATP-linked respiration), oligomycin (1 μM)-treated proton leak, and FCCP (1 μM)-treated maximal OCR, which includes spare capacity, were monitored. Antimycin A (0.5 μM) was used to confirm the specificity for mitochondrial OCR. ECAR at basal OCR condition was monitored and quantified.

### 4.3. Measurement of Extracellular Lactate and Glucose

The concentration of lactate and glucose in cultured medium was measured using a YSI 7100 Bioanalyzer (Yellow Springs, OH, USA). Briefly, cells (4 × 10^4^) were seeded on 12-well plates and cultured for 24 h. After replenishing with fresh medium for 2 days, the conditioned medium was harvested and applied to the chamber assembled with a glucose/lactate membrane kit (cat. #, YSI 2365 and YSI 2329) to measure simultaneously glucose and lactate concentrations. The concentration values were obtained by comparing with standard calibration curves for lactate and glucose concentrations and were normalized by total cell lysate protein.

### 4.4. Measurement of Cellular ATP Level

Intracellular ATP level was measured using an ATPlite^TM^ Luminescence Assay System (PerkinElmer, Waltham, MA, USA) according to provided instructions. In brief, cells (1 × 10^4^) were lysised with mammalian cell lysis solution (150 μL). Cell lysate (150 μL) was mixed with substrate solution (50 μL), and the luciferase activity was analyzed using BioTek microplate reader (Synergy H1, Winooski, VT, USA).

### 4.5. Western Blot Analysis

Western blotting was performed using a standard procedure [41]. Cell lysates (15 μg) were separated on 8–12% SDS-polyacrylamide gels and then transferred onto nitrocellulose membrane (Amersham Biosciences Protran^TM^, GE Healthcare, Illinois, CA, USA). Then, the membranes were incubated overnight with the following primary antibodies. Antibodies for NFE2L1 (#8052,1:1000), phospho-STAT3 (#9131, 1:1000), and EphB2 (#83029, 1:1000) were purchased from Cell Signaling Technology (Danvers, MA, USA). Antibodies for STAT3 (GTX15789, 1:1000), USF2 (GTX129239, 1:1000), and ENO2 (GTX10169, 1:1000) antibodies were purchased from GenTex, Inc. (Irvine, CA, USA). Antibodies for β-actin (sc-1616, 1:3000) and c-JUN (sc-1694,1:1000) were obtained from Santa Cruz Biotechnology, Inc. (Dallas, TX, USA). Antibodies against serpinE1 (NBP1-19773, 1:1000), STX12 (PA5-59407, 1:500), SREBP1 (557036,1:1000), and α-tubulin (05–829, 1:10000) were purchased from Novus Biologicals (Centennial, CO, USA), Thermo Fisher Scientific (Waltham, MA, USA), BD Biosciences (San Jose, CA, USA) and Millipore (Billerica, MA, USA), respectively. NDUFA9 antibody (A21344, 1:1000) was obtained from Molecular Probes (Eugene, OR USA). Then, the membranes were incubated with horseradish peroxidase-conjugated anti-mouse (sc-2005, 1:5000) or anti-rabbit (sc-2004, 1:3000) antibody (Santa Cruz Biotechnology) for 1–2 h. Immunoreactive proteins were visualized using an enhanced chemiluminescent (ECL) system (West Save, AbFrontier, Seoul, Korea) and X-Ray film (Agfa, Mortsel, Belgium) or ChemiDoc Touch Imaging System (Bio-Rad, Hercules, CA, USA).

### 4.6. qRT-PCR

Total RNA was isolated using the NucleoSpin^TM^ RNA Plus kit (MACHEREY-NAGEL GmbH & Co. KG, Düren, Germany), according to the instruction provided. After generating primary cDNAs using avian myeloblastosis virus reverse transcriptase (Promega, Madison, WI, USA), PCR reaction was performed using the THUNDERBIRD SYBRTM qPCR Mix (Toyobo Co., Ltd., Osaka, Japan) and primer sets for target genes. The PCR primer sets were produced by Macrogen, Inc. (Seoul, Korea) as follows: NFE2L1 (5′-CTGCTAGTGGATGGAGAGA and 5′-GCTTCTGTTATGCTGGAAATG), STAT3 (5′-CGATGCTGGAGGAGAGAATC and 5′-GACCAGCAACCTGACTTTAG), STX12 (5′-GGCAATGTGGAAAGCTCAGAGG and 5′-TCACTGACAGGACAAGCACCAG), ENO2 (5′-CAAGGACAAATACGGCAAGG and 5′-GTATCGGGAAGGATCAGTGG), SerpinE1 (5′-CTCATCAGCCACTGGAAAGGCA and 5′-GACTCGTGAAGTCAGCCTGAAAC), EphB2 (5′-ATCTATGTCTTCCAGGTGCG and 5′-TCAAACCCCCGTCTGTTAC), TBP (5′-CACCTTACGCTCAGGGCTT and 5′-CTGAATAGGCTGTGGGGTCA), NDUFA9 (5′-CGAGACTGGGAAACCAAAAAC and 5′-GCATCCGCTCCACTTTATCC), and β-actin (5′-CCTTCCTGGGCATGGAGTCCTGT and 5′-GGAGCAATGATCTTGATCTTC).

### 4.7. Cell Invasion Assay

Cell invasion assay was performed by using Transwell^TM^ Permeable Supports (8 µm pore size; Corning, Acton, MA, USA), which were precoated with 7% Growth Factor Reduced BD Matrigel^TM^ Matrix [23]. Cells underwent serum starvation for 6 to 16 h and then 2 × 10^4^ cells in 100 μL serum-free media were seeded in the upper chamber. The lower chamber was filled with full media containing 10% fetal bovine serum. After incubation at 37 °C for 48 h, invaded cells were stained with Hematoxylin (Sigma, Saint Louis, MO, USA) and Eosin Y (Sigma) solution and counted by using Image J software. All experiments were performed in at least three independent experiments.

### 4.8. Measurement of Intracellular and Mitochondrial ROS Level

To determine intracellular ROS levels, a dichlorofluorescin diacetate fluorescent probe (DCFH-DA; Molecular Probes) was used. A MitoSOX^TM^ (Molecular Probes) fluorescent probe was used for the mitochondrial ROS level. Briefly, cells were incubated in media containing 20 µM DCFH-DA or 2.5 µM MitoSOX^TM^ for 20 min at 37 °C. Stained cells were washed and resuspended in PBS and then analyzed by flow cytometry (FACSCanto^TM^ II, BD Biosciences). Mean values of arbitrary fluorescence unit for 10,000 cells were obtained and expressed as percentage of control.

### 4.9. Immunocytochemistry

Cells were fixed with fixation solution (Methanol: Acetone = 1:1) at −20 °C for 10 min. Primary NFE2L1 antibody (Cell Signaling Technology, #8052) and Alexa Fluor 488-conjugated anti-rabbit secondary antibody (Invitrogen, A11059) were used to detect NFE2L1. Nuclei were counter-stained with DAPI (Sigma), and the stained cells was visualized with a Confocal laser scanning microscope (K1-Fluo, Nanoscope Systems. Daejeon, Korea).

### 4.10. Construction of STAT3 Mutant, STX12, and NFE2L1 Expression Plasmids

The STAT3-dominant negative mutant plasmid, pcDNA-STAT3-Y705F, was generated by performing site-directed mutagenesis to the pcDNA3.1-STAT3-HA plasmid [42] using a primer set: 5′-TAGCGCTGCCCCATTCCTGAAG and 5′- ATAAACTTGGTCTTCAGGAATGGGGCA. To generate STX12 lentiviral plasmid, STX12 cDNA was amplified by targeted PCR against total cDNAs of Ch-L using a primer set, 5′-AAGCTAGCACCATGTCATACGGTCC and 5′-CGCGGATCCTCACTTCGTTTTATAAAC. The amplified PCR product was inserted between NheI and BamHI sites of the Lenti-CMV-GFP-2A-Puro Vector (LV073, Applied Biological Materials Inc., Richmond, BC, Canada). To construct the pcDNA-NFE2L1 expression plasmid, NFE2L1 cDNA was amplified by PCR against the lentiviral NFE2L1 plasmid (LV237920, Applied Biological Materials Inc.) using a primer set (5′-GCAAGCTTCCATGCTTTCTCTGAAG and 5′- GTCTAGACTTTCTCCGGTCCTT) and subcloned into pcDNA3.1-V5-His-TOPO (Invitrogen) by the TA cloning method. All the primer sets for cloning were generated by Macrogen Inc.

### 4.11. Transfection of cDNA Plasmids and siRNAs

To introduce plasmids and small interfering RNAs (siRNAs) into cells, cells were transfected with the plasmids and target siRNA duplexes using FuGENE^®^ HD (Promega) and Oligofectamine^TM^ Reagent (Invitrogen), respectively, according to the instructions provided. Target siRNAs used in this study were obtained from Bioneer (Daejeon, Korea) as follows: AHR, 5′-CACUCAGACUACCACACAU; Sp1, 5′-CAGAUACCAGACCUCUUCU; TCF1, 5′-GUGAUGAGCUACCAACCAA; STAT1, 5′-CUGACUUCCAUGCGGUUGA; YY1, 5′-UGAAGCUCACCUGUUGCUU; RXRα, 5′-GGUGGGACAAUCUUUAAUU; CEBPδ, 5′-CGAGAGAAGCUAAACGUGU; USF2, 5′-CCUCCACUUGGAAACGGUA (#1) and 5′-CAGGAACACAGAGGACGAU (#2); NDUFA9, 5′-AAUCAUACCCUAUCGGUGU; STAT3, 5′-UGUUCUCUGAGACCCAUGA (#1) and 5′-CUAUCUAAGCCCUAGGUUU (#2); c-JUN, 5′-ACUGUAGAUUGCUUCUGUA (#1) and 5′-GAACUAAAGCCAAGGGUAU (#2); SREBP1, 5′-CCACCGUUUCUUCGUGGAU (#1) and 5′-GACUGUCAGCAGAUGCUCA (#2); STX12, 5′-GCUGGUCAGUUACUGGAGU (#1) and 5′-GUGUCAUCCACAGCAGUUC (#2); NFE2L1, 5′-ACUGUAGAUUGCUUCUGUA (#1) and 5′-GAACUAAAGCCAAGGGUAU (#2); negative control, 5′-CCUACGCCACCAAUUUCGU.

### 4.12. Construction of Promoter Reporter Plasmid and Promoter Assay

The 1899 bp human NFE2L1 promoter DNA fragment (−1580 to +318, NC_000017.11) was amplified by polymerase chain reaction (PCR) using the total genomic DNA of Ch-L cell and a primer set, 5′-CTCGAGCATCCTCCAAGGTGTGGT and 5′-AAGCTTAGCCAGCTTTCCTCTCGG (Macrogen, Inc., Seoul, Korea). The amplified NFE2L1 promoter DNA fragment was inserted between the XhoI and HindIII sites of the pGL3-basic plasmid (Promega). The 443 bp promoter region (−124 to +318) was amplified against the 1899 bp promoter plasmid using a primer set, 5′-GATTCTCGAGATGAGGAGGTAC and 5′-AAGCTTAGCCAGCTTTCCTCTCGG (Macrogen, Inc). The amplified NFE2L1 promoter DNA fragment (443 bp) was also inserted between the XhoI and HindIII sites of the pGL3-basic plasmid (Promega). After construction, DNA sequences of the inserted DNA fragments were confirmed (Cosmogenetech Inc., Seoul, Korea).

To monitor NFE2L1 promoter activity, cells were transfected with a total of 1 µg of plasmid (950 ng of pGL3-NFE2L1 promoter plasmid and 50 ng of renilla luciferase plasmid) using FuGENE^®^ HD reagent (Promega). After 48 h, the luciferase activity of cell lysate (15 μg) was measured using the Dual-Luciferase^TM^ reporter assay system (Promega) and BioTek microplate reader (Synergy H1) according to the protocol provided. The promoter-annealed firefly luciferase activity was normalized by the renilla luciferase activity, which was used as an internal control.

### 4.13. Production of Recombinant Lentiviruses and Generation of Stably Target Expressing Cell Clones

A LentiCRISPR-v2 plasmid containing NFE2L1 single guide RNA (sgNFE2L1) was constructed according to the methods described previously [43]. Briefly, an annealed complimentary oligonucleotide for the sgNFE2L1 sequence was inserted to the BsmBI-digested LentiCRISPRv2 plasmid (Addgene, #83480, Watertown, MA, USA). The oligonucleotide for sgNFE2L1 (5′-GAAAGGGATCTTCATGGCTC) was synthesized by Macrogen, Inc. Undigested LentiCRISPR-v2 plasmid lacking any sgRNA was used as a negative control (sgNC).

To generate the recombinant lentiviruses harboring sgNFE2L1, shNFE2L1, or recombinant STX12 cDNA, a 293T packaging cell was transfected with LentiCRISPR-sgNFE2L1 plasmid, pLKO.1-puro plasmid [22], or lenti-STX12 plasmid using Lipofectamine^TM^ (Invitrogen). Medium containing recombinant lentivirus was harvested 3 days after transfection and filtered through a 0.45 μm filter unit (UFC 920008, Millipore). A filtered medium was mixed with polybrene (8 μg/mL, Sigma-Aldrich) for facilitated infection and stored at −80 °C. After cells were infected with the recombinant lentiviruses, cell clones stably expressing target gene were selected with 30 µg/mL blasticidin (Enzo Life Sciences, Farmingdale, NY, USA) for sgNFE2L1 expression and with 3 µg/mL puromycin (Sigma) for both shNFE2L1 and STX12 expression.

### 4.14. Microarray Transcriptome Profiling of Hepatoma Cell Lines

Total RNA isolation, labeling, and hybridizations on an Affymetrix GeneChip Human 2.0 ST array were performed according to the manufacturer’s protocol (GeneChip Whole Transcript PLUS reagent Kit, Affymetrix, Inc., Santa Clara, CA, USA). Signal values were computed using the Affymetrix^®^ GeneChip™ Command Console software. All microarray data were deposited into the Gene Expression Omnibus under the accession number of GSE137054. Raw data were extracted automatically in Affymetrix data extraction protocol using the software provided by Affymetrix GeneChip^®^ Command Console^®^ Software (AGCC). After importing CEL files, the data were summarized and normalized with the robust multi-average (RMA) method implemented in Affymetrix^®^ Expression Console™ Software (EC). We performed further analysis using the gene-level RMA processed data.

### 4.15. Analysis of RNA-seq Data of TCGA_LIHC

We used the RNAseq data of the TCGA-LIHC cohort (*n* = 424; 371 primary tumors, three recurrent tumors, and 50 solid normal liver tissues), which were obtained from the NCI’s Genomic Data Commons portal (https://portal.gdc.cancer.gov/). RNA abundance was estimated as log2 transformed value, log_2_(FPKM+1), and quantile-normalized. Association between two gene expressions in primary tumors was determined by performing Pearson’s correlation test. To calculate the enrichment of EMT-related genes, preRanked GSEA [44] was performed using the EMT core gene signature (EMT_UP; *n* = 91), which was previously identified by Taube et al. [45]. To compare the enrichment of EMT score based on the NFE2L1, STX12, and/or NDUFA9 expression level, we divided the primary tumors into high and low groups for each target gene, which showed an upper and lower quantile of target expression, respectively. The overall survival (OS) time of two groups was compared by Kaplan–Meier (KM) survival curves, and its statistical significance was estimated by the Cox–Mantel log-rank test. All statistics were performed using the computing environment R (R version 3.3.2).

### 4.16. Immunohistochemistry of HCC Tissue Microarray Samples

Immunohistochemistry was performed with the standard protocol [46]. Briefly, paraffin-embedded tissue slides of human HCC tissue microarray (TMA) (NBP2-30221, Novus Biologicals) were deparaffinized and rehydrated. Endogenous peroxidase activity was blocked with 3% hydrogen peroxide, and antigen retrieval was performed in 10 mM citrate buffer. The primary antibodies for NFE2L1 (#8052, Cell Signaling Technology) and STX12 (PA5-59407, Thermo Fisher Scientific) were used. The signals were developed with a diaminobenzidine (DAB) substrate solution (Thermo Fisher Scientific), and nuclei were counter-stained with hematoxylin. Images were captured using the slide scanner AxioScan. ZI (Zeiss, Oberkochen, Germany). DAB-stained intensities were graded 0 (no staining), 1 (weak), 2 (moderate), or 3 (strong) for NFE2L1 and 0 (no staining), 1 (weak), or 2 (moderate) for STX12.

### 4.17. Statistical Analysis

All assays with cell line experiments were performed at least thrice in triplicate. The values were expressed as average ± standard deviation (SD). Comparisons between groups were estimated using the Student *t*-test. A *p*-value < 0.05 was considered statistically significant.

## 5. Conclusions

In conclusion, we present a novel mitochondrial dysfunction-mediated retrograde signaling pathway, mitochondrial ROS-mediated STAT3 activation, and resulting transcriptomic reprogramming in liver cancer progression, providing the NDUFA9/NFE2L1/STX12 axis as a key prognostic marker of aggressive liver cancer with mitochondrial defect.

## Figures and Tables

**Figure 1 cancers-12-02632-f001:**
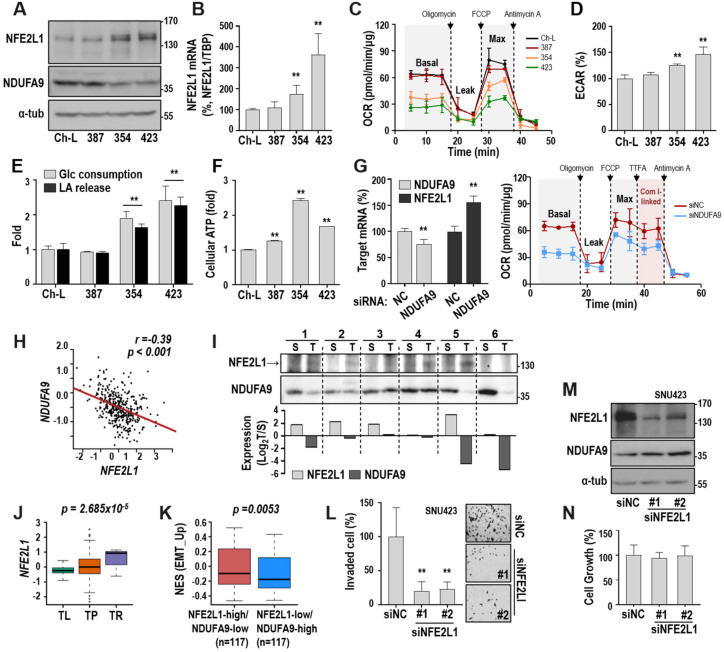
Nuclear factor-erythroid 2 like 1 (NFE2L1) expression is associated with decreased mitochondrial activity and involved in hepatoma cell invasiveness. (**A**–**F**) SNU hepatoma cell lines (SNU387, SNU354, and SNU423) and the Ch-L clone were cultured for 2 days to maintain an exponentially growing state. (**A**) Western blot. (Appendix A) (**B**) qRT-PCR. NFE2L1 mRNA level was normalized by TBP mRNA level. ** *p* < 0.01 vs. Ch-L by the Student *t*-test. (**C**) Cellular OCR was monitored by using XF-24 Extracellular Flux Analyzer as described in ‘Materials and Methods’. (**D**) Extracellular acidification rate (ECAR) at the basal cell respiration condition was measured by the XF-24 analyzer. ** *p* < 0.01 vs. Ch-L. (**E**) Extracellular lactate (LA) release and glucose (Glc) consumption were monitored as described in ‘Materials and Methods’. ** *p* < 0.01 vs. Ch-L. (**F**) Cellular ATP level was analyzed by using a luminescence-linked ATP assay kit as described in ‘Materials and Methods’. **, *p* < 0.01 vs. Ch-L. (**G**) Ch-L was transfected with siRNA for NDUFA9 (siNDUFA9) for 72 h. NFE2L1 and NDUFA9 mRNA level by qRT-PCR (left) and cellular OCR level (right). (**H**) Association analysis of NFE2L1 and NDUFA9 in the 371 The Cancer Genome Atlas Liver Hepatocellular carcinoma (TCGA_LIHC) tumors by Pearson’s product moment correlation test. (**I**) Tumor and surrounding tissue samples of six human HCC tissues were subjected to Western blot (Appendix A) analysis (upper). Quantification is shown as log2 value of the ratio (T/S) of tumor to surrounding sample intensities (lower). (**J**,**K**) Bioinformatics analysis from TCGA_LIHC dataset. (**J**) Boxplot shows NFE2L1 expression level in different types of tissue. TL (*n* = 50), TP (*n* = 371), and TR (*n* = 3) indicate tissues from normal liver, primary tumor, and recurrent tumor, respectively. P-values from Welch two-sample t-test (TP vs. TL) are indicated. (**K**) Boxplot of the normalized enrichment score (NES) for epithelial–mesenchymal transition (EMT) signature of the primary HCC tissues from the TCGA_LIHC dataset. Bottom, middle, and top lines of each box indicate the first quartile, median, and third quartile values, respectively. Whiskers represent the minimum and maximum values. For comparison, primary tumors were divided into two groups based on the median level of NFE2L1 and NDUFA9 expression level. *p*-value from Welch two-sample t-test is indicated. (**L**–**N**) SNU423 was transfected with two different siRNAs for NFE2L1 (siNFE2L1 #1 and #2) for 72 h. Nonspecific siRNA was used as negative control (siNC). (**L**) Cell invasion assay was performed by using a Matrigel-coated Transwell system as described in the ‘Materials and Methods’. Invaded cells were counted and quantified. Representative invaded cell images are shown in the right panel. ** *p* < 0.01 vs. siNC by the Student t-test. (**M**) Western blot analysis. (Appendix A). (**N**) Cell growth was monitored by counting trypan-blue negative viable cells. No significant dead cells were observed.

**Figure 2 cancers-12-02632-f002:**
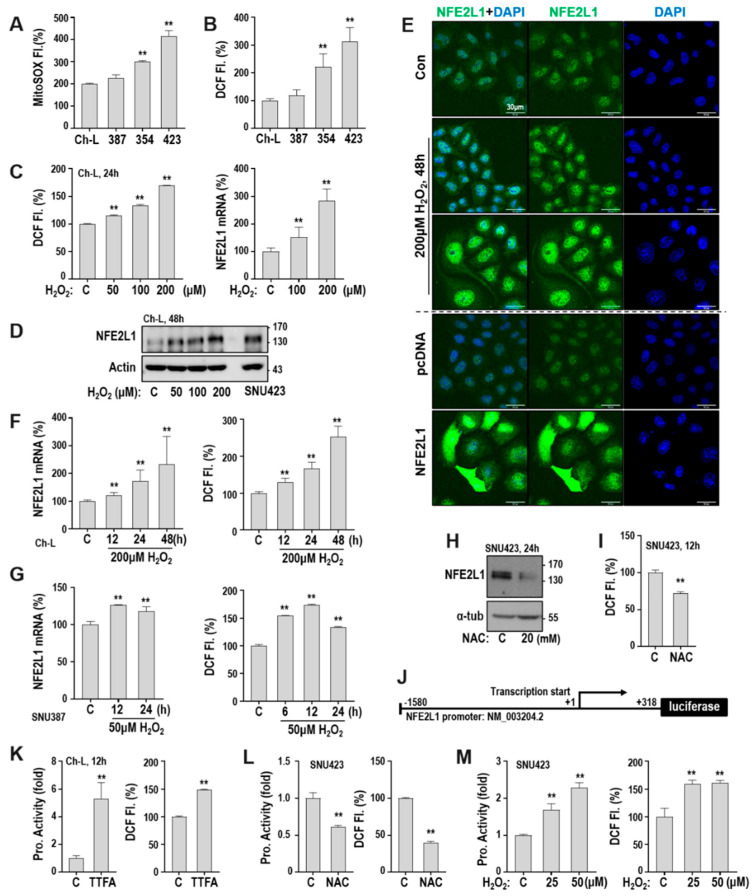
Mitochondrial defect increases NFE2L1 transcription via reactive oxygen species (ROS). (**A, B**) Mitochondrial ROS (**A**) and cellular ROS (**B**) levels were monitored by flowcytometric analysis after staining cells with MitoSOX and DCFH-DA, respectively. ** *p* < 0.01 vs. Ch-L by the Student t-test. (**C**–**F**). Ch-L was treated with the indicated concentrations of H_2_O_2_ for the indicated time periods. ** *p* < 0.01 vs. C by the Student t-test. (**C**) Cellular ROS level was monitored by using DCFH-DA (left panel), and the NFE2L1 mRNA level was examined by qRT-PCR (right). (**D**) Western blot. (Appendix A) SNU423 was used as a control of NFE2L1 level. (**E**) Immunocytochemistry for the intracellular localization of NFE2L1. Nucleus was stained with DAPI. NFE2L1 overexpressed by the transfection of NFE2L1-V5 plasmid was used as a control. (**F**) Time-course NFE2L1 expression in response to H_2_O_2_. NFE2L1 mRNA level by qRT-PCR (left) and cellular ROS level (right). (**G**) SNU387 was treated with 50 μM H_2_O_2_ for the indicated time periods. Lower H_2_O_2_ concentrations were applied to SNU387 due to the cell’s different sensitivity to H_2_O_2_ from Ch-L. Over 100 μM H_2_O_2_ induced the cell death of SNU387. NFE2L1 mRNA level (left panel) and cellular ROS level (right panel). ** *p* < 0.01 vs. C by the Student *t*-test. (**H,I**) SNU423 was treated with 20 mM N-acetyl cysteine (NAC) for 24 h. (**H**) Western blot. (Appendix A) (**I**) Cellular ROS level. (**J**) Schematic design of NFE2L1-pGL3 reporter plasmid containing NFE2L1 promoter region from −1580 to +318. (**K**) Ch-L was transfected with NFE2L1-pGL3 plasmid together with renilla luciferase plasmid for 48 h and then exposed to 400 μM TTFA (a complex II inhibitor) for 12 h. Promoter activity (left) and cellular ROS level (right). (**L**,**M**) SNU423 was transfected with NFE2L1-pGL3 plasmid and renilla luciferase plasmid for 48 h and then treated with 20 mM NAC (**L**) for 24 h or with the indicated concentrations of H_2_O_2_ for 48 h (**M**). Promoter activity (left) and cellular ROS level (right) were monitored. ** *p* < 0.01 vs. C by the Student *t*-test.

**Figure 3 cancers-12-02632-f003:**
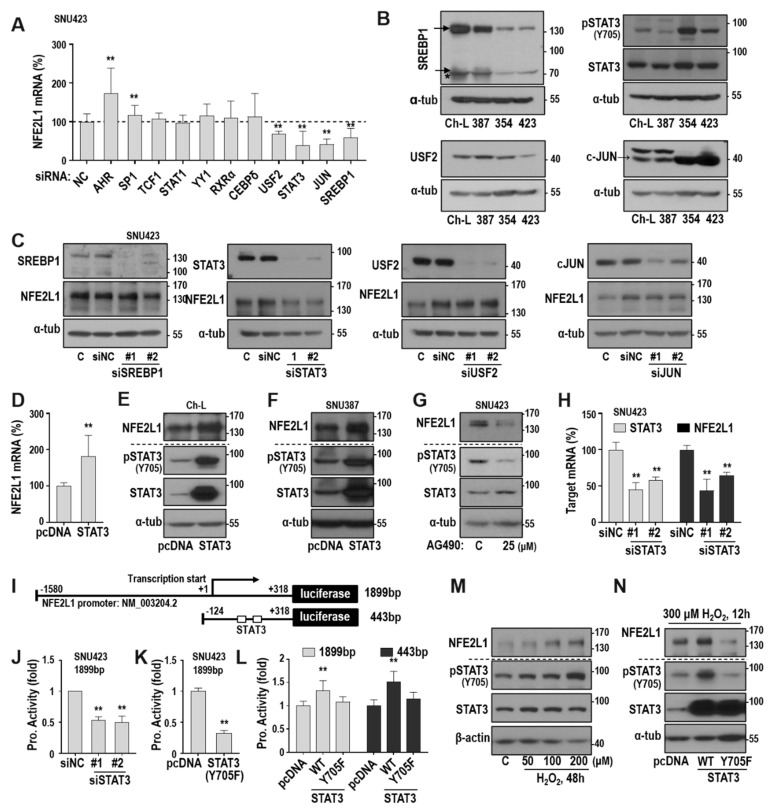
ROS-mediated STAT3 activation is the upstream regulator of NFE2L1 expression. (**A**) After introducing siRNAs for the 11 TFs individually to SNU423, the NFE2L1 mRNA level was monitored by qRT-PCR. ** *p* < 0.01 vs. siNC by the Student *t*-test. (**B**) Western blot (Appendix A) analysis for protein levels of four selected TFs (USF2, c-Jun, SREBP1, and STAT3) and phosphorylated status on tyrosine 705 of STAT3 (pSTAT3, Y705). *, The truncated active form of SREBP1 was also detected. (**C**) Western blot analysis (Appendix A) after siRNAs for the four TFs were individually introduced into SNU423 for 72 h. #1 and #2 indicate two different siRNAs for individual target genes. (**D**,**E**) Ch-L was transfected with pcDNA3-STAT3-HA plasmid for 48 h and subjected to qRT-PCR (**D**) and Western blot analysis (**E**) (Appendix A). ** *p* < 0.01 vs. mock transfection with pcDNA3 plasmid (pcDNA) by the Student t-test. (**F**) SNU387 was transfected with pcDNA3-STAT3-HA for 48 h and subjected to Western blot analysis. (Appendix A) (**G**) SNU423 was treated with 25μM AG490 (a STAT3 inhibitor) for 24 h and subjected to Western blot analysis. (Appendix A) (**H**) SNU423 was transfected with siSTAT3 for 72 h and qRT-PCR was performed to monitor NFE2L1 and STAT3 mRNA level. ** *p* < 0.01 vs. siNC by the Student t-test. (**I**) Schematic diagram of NFE2L1-pGL3 reporter plasmids with the indicated promoter length (1899 bp or 443 bp). The open square boxes indicate two putative STAT3 binding sites located in +67 to +80 and +174 to +189 bps. (**J**) Promoter assay was performed after SNU423 was transfected with siSTAT3 for 12h and then with the NFE2L1 promoter reporter plasmid (1899 bp) for 48 h. (**K**) SNU423 was transfected with the 1899 bp promoter reporter plasmid together with pcDNA–STAT3–Y705F dominant negative mutant plasmid and for 48 h and then subjected to promoter assay. (**L**) Promoter assay was performed after Ch-L was transfected with the two NFE2L1 promoter reporter plasmids, 1899 bp (left) and 443 bp (right), together with pcDNA-STAT3 or pcDNA-STAT3-Y705F mutant plasmid for 48 h. (**M**) Ch-L was treated with the indicated concentrations of H_2_O_2_ for 48 h and subjected to Western blot analysis. (Appendix A) (**N**) Ch-L was transfected with STAT3 WT or mutant plasmids for 48 h and then exposed to 300 µM H_2_O_2_ for 12 h. Western blot analysis (Appendix A) was performed. In J to L, ** *p* < 0.01 vs. pcDNA or siNC by the Student t-test.

**Figure 4 cancers-12-02632-f004:**
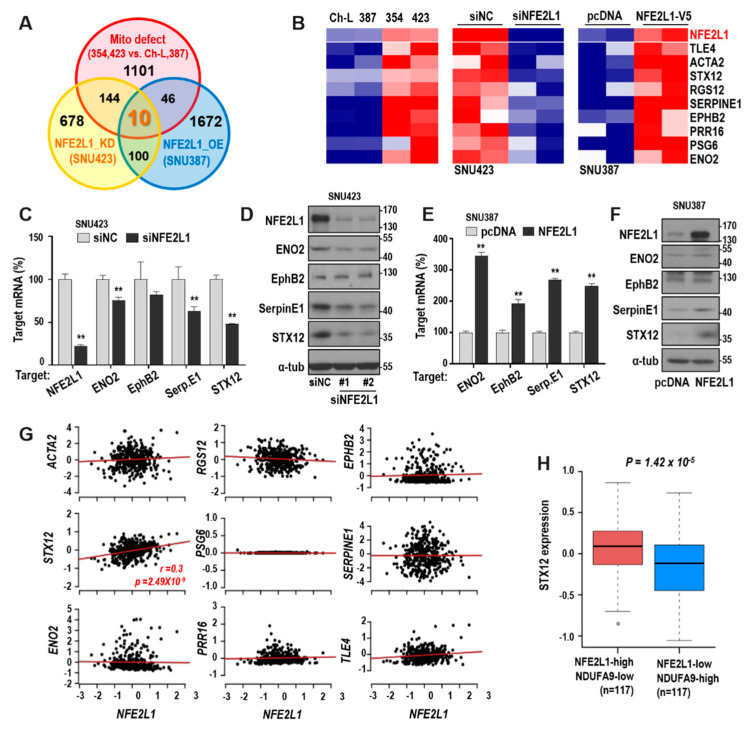
Identification of downstream target genes commonly upregulated by mitochondrial defect and NFE2L1. Transcriptome data of four cell lines (Ch-L and SNU387 harboring active mitochondria, and SNU354 and SNU423 harboring defective mitochondria) were obtained from our previous report [22]. Two independent sets of SNU387 cells transfected with pcDNA-NFE2L1 plasmid (NFE2L1_OE) for 48 h and of SNU423 cells with siNFE2L1 (NFE2L1_KD) for 72 h were applied to cDNA microarray for transcriptome profiling. (**A**) Venn diagram of commonly upregulated genes with mitochondrial defect (Mito defect, SNU354 and SNU423 vs. Ch-L and SNU387) and NFE2L1-dependent regulation. (**B**) Heatmaps show the expression of 10 commonly upregulated genes in four hepatoma cell lines (left), NFE2L1-depleted SNU423 cells (middle), and NFE2L1-overexpressed SNU387 cells (right). Each column and row represent independent samples and the indicated genes, respectively. Red and blue color indicates the high and low expression, respectively. (**C**,**E**) Validation of four target mRNA levels by qRT-PCR. (**D**,**F**) Validation of four target protein levels by Western blot analysis. #1 and #2 indicate two different siRNAs for NFE2L1. (Appendix A) ** *p* < 0.01 vs. pcDNA or siNC by the Student *t*-test. (**G**) Associations of NFE2L1 with the 10 commonly upregulated genes were evaluated using the TCGA-LIHC cohort (*n* = 371). Pearson’s product moment correlation test was performed. The correlation estimate and the p-value with statistical significance are marked. (**H**) Boxplot for STX12 expression level of primary HCC tissues from the TCGA-LIHC cohort. Bottom, middle, and top lines of each box indicate the first quartile, median, and third quartile values, respectively. Whiskers represent the minimum and maximum values. For comparison, primary tumors were divided into two groups, based on the median expression level of NFE2L1 and NDUFA9: a group with both above the median NFE2L1 level and below the median NDUFA9 level, a group with both below the median NFE2L1 level and above the median NDUFA9 level. *p*-value from Wilcoxon t-test is indicated.

**Figure 5 cancers-12-02632-f005:**
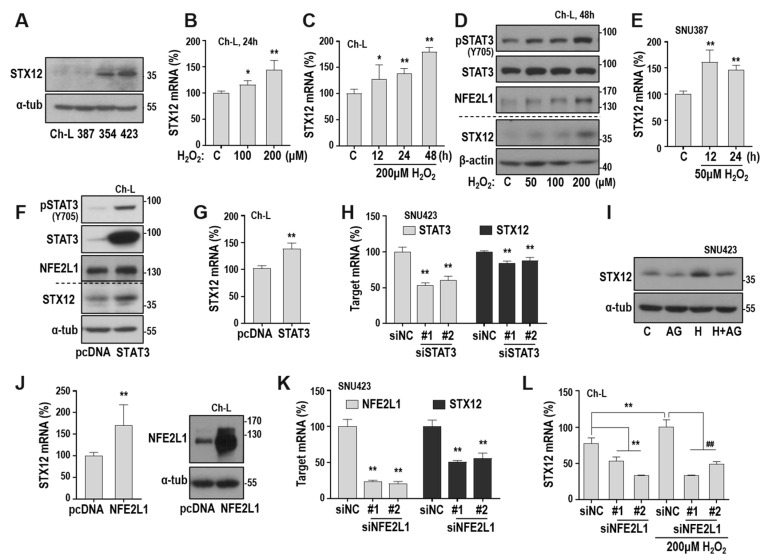
STX12 is a key downstream target of the ROS/STAT3/NFE2L1 axis. (**A**) Western blotting. (Appendix A) (**B**–**E**) Ch-L and SNU387 cells were cultured with different concentrations of H_2_O_2_. Validation of STX12 mRNA level by qRT-PCR in Ch-L (**B**,**C**) and SNU387 (**E**) cells. ** *p* < 0.01; * *p* < 0.05 vs. C by the Student t-test. (**D**) Western blot analysis. (Appendix A) (**F**,**G**) Ch-L cell was transfected with pcDNA3-STAT3 plasmid (STAT3) for 48 h and then Western blot analysis (**F**) (Appendix A) and qRT-PCR (**G**) were performed. (**H**) SNU423 was transfected with siSTAT3 for 72 h and subjected to qRT-PCR. ** *p* < 0.01 vs. siNC or pcDNA by the Student *t*-test. #1 and #2 indicate two different siRNAs for STAT3. (**I**) SNU423 was exposed to 50 μM H_2_O_2_ with or without the pretreatment of 25 µM AG490 (AG), and then Western blot analysis was performed. (Appendix A) (**J**) Ch-L cell was transfected with pcDNA3-NFE2L1 plasmid (NFE2L1) for 48 h and subjected to qRT-PCR (left) and Western blot analysis (right) (Appendix A). (**K**) SNU423 was transfected with siNFE2L1 for 72 h and subjected to qRT-PCR. ** *p* < 0.01 vs. siNC by the Student t-test. #1 and #2 indicate two different siRNAs for NFE2L1. (**L**) Ch-L was transfected with siNFE2L1 for 24 h and then exposed to 200 µM H_2_O_2_ for 48 h. STX12 mRNA level was monitored by qRT-PCR. ** *p* < 0.01 vs. siNC without H_2_O_2_; ^##^
*p*<0.01 vs. siNC with H_2_O_2_ by the Student *t*-test. #1 and #2 indicate two different siRNAs for NFE2L1.

**Figure 6 cancers-12-02632-f006:**
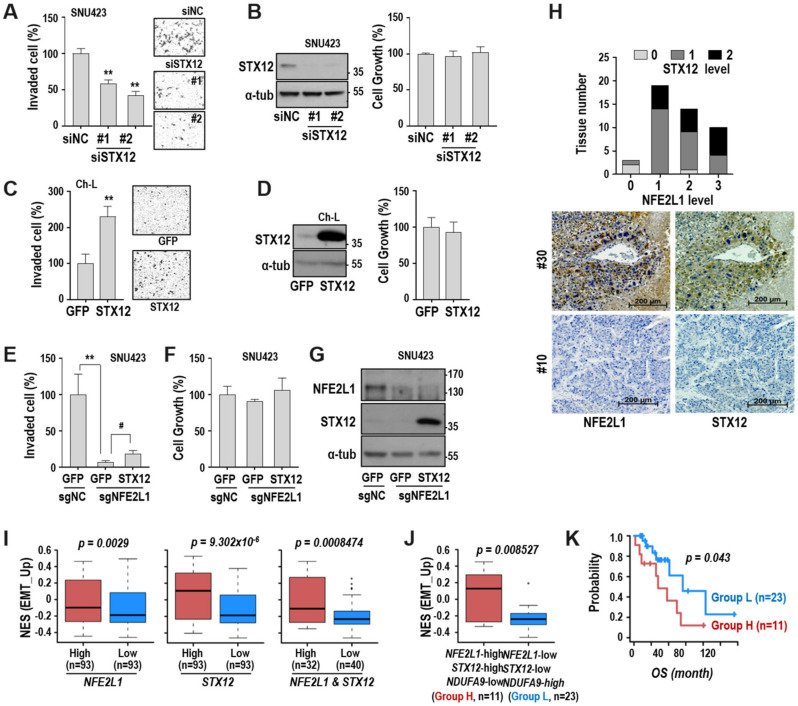
STX12 is a key regulator of hepatoma cell invasiveness. (**A**,**B**) SNU423 cell was transfected with STX12 siRNA for 72 h. (**A**) Cell invasion assay using a Matrigel-coated Transwell system. Representative invaded cell images are shown in the right panel. ** *p* < 0.01 vs. siNC by the Student t-test. (**B**) Western blot (left) (Appendix A) and cell growth rate (right). #1 and #2 indicate two different siRNAs for STX12. (**C**,**D**) Ch-L was transfected with STX12 plasmid for 48 h. (**C**) Cell invasion assay. Representative invaded cell images are shown in the right panel. ** *p* < 0.01 vs. GFP by the Student t-test. (**D**) Western blot (left) (Appendix A) and cell growth rate (right). (**E**–**G**) SNU423 clone, which was stably suppressing NFE2L1 with sgNFE2L1 RNA, was further infected with lentivirus harboring STX12 or GFP. (**E**) Cell invasion assay. ** *p* < 0.01 vs. sgNC; ^#^ < 0.05 vs. GFP by the Student *t*-test. (**F**) Cell growth. (**G**) Western blot. (Appendix A) (**H**) Immunohistochemistry of HCC tissue microarray, as described in ‘Materials and Methods’. Lower panel shows representative images for NFE2L1 and STX12 immunostaining. (**I**,**J**) Boxplots of the normalized enrichment score (NES) for EMT signature of the primary HCC tissues from the TCGA_LIHC dataset. Bottom, middle, and top lines of each box indicate the first quartile, median, and third quartile values, respectively. Whiskers represent the minimum and maximum values. *p*-values from Welch two-sample *t*-test are indicated. (**I**) Tumor samples were divided into high and low groups, according to the expression level of NFE2L1 (left) or STX12 (middle), respectively, as described in ‘Materials and Methods’. To evaluate the effect of co-expression on the EMT signature, tumor samples were divided into two groups, based on the co-expression level of both NFE2L1 and STX12 (right). (**J**) The NFE2L1 and STX12 co-expressing group (right panel of Figure 6I) was further subdivided into an H (*n* = 11) and L group (*n* = 23) according to the NDUFA9 expression level. (**K**) Overall survival (OS) time of the H and L group was compared based on the Kaplan–Meier survival analysis. Statistical significance for KM survival was estimated by the Cox–Mantel log-rank test.

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
