# Peer review of "Mitochondrial Respiratory Defect Enhances Hepatoma Cell Invasiveness via STAT3/NFE2L1/STX12 Axis"

_cancers, 2020, doi:10.3390/cancers12092632_

Round 1

Reviewer 1 Report

Mitochondrial respiratory defect enhances hepatoma 2 cell invasiveness via STAT3/NFE2L1/STX12 axis

Young-Kyoung Lee, So Mee Kwon, Eun-beom Lee, Gyeong-Hyeon Kim, Seongki Min, Sun-Mi Hong, Hee-Jung Wang, Dong Min Lee1, Kyeong Sook Choi, Tae Jun Park, Gyesoon Yoon

In this manuscript, Lee et al. explore the role of mitochondrial OXPHOS defect in the induction of Nuclear factor-erythroid 2 like 1 (NFE2L1) expression via reactive oxygen species (ROS)-mediated STAT3 activation and the upregulation of NFE2L1 increased hepatoma cell invasiveness by inducing syntaxin 12 (STX12) expression. This study is well designed, and the findings are clear. Overall, this paper is well-written and technically sound and would be suitable for publication in this journal. However, some additional evidence is needed to strengthen the conclusion of this manuscript:

Major comment

  1. In Figure 1C and 1D, what authors mean by % OCR, is it basal or maximal? Please add OCR trace in this figure and quantify, basal, maximum, ATP-linked respiration, spare capacity, and proton leak separately. Also, what about the ECAR activity in the same groups.

Minor comments

  1. In figure 2A, please explain how this data was quantified. The graph represent MitoSoxFl %, however, control group Ch-L shows a value of 200%.
  2. Figure 3C, text “NFE2L1” in the second blot overlaps with 130kd in the first blot, please fix it.
  3. Most of the methods lacks citation.
  4. In methods section 4.3., Western blot analysis, please add information on antibodies dilution, secondary antibodies, and developing methods.
  5. The statistic is poorly described. No information on tests, N numbers, and significant value is described.

Author Response

<Response to reviewer 1>

In this manuscript, Lee et al. explore the role of mitochondrial OXPHOS defect in the induction of Nuclear factor-erythroid 2 like 1 (NFE2L1) expression via reactive oxygen species (ROS)-mediated STAT3 activation and the upregulation of NFE2L1 increased hepatoma cell invasiveness by inducing syntaxin 12 (STX12) expression. This study is well designed, and the findings are clear. Overall, this paper is well-written and technically sound and would be suitable for publication in this journal. However, some additional evidence is needed to strengthen the conclusion of this manuscript:

Response: We are sincerely grateful to the reviewer for the critical and helpful comments.

Major comment

1. In Figure 1C and 1D, what authors mean by % OCR, is it basal or maximal? Please add OCR trace in this figure and quantify, basal, maximum, ATP-linked respiration, spare capacity, and proton leak separately. Also, what about the ECAR activity in the same groups.

Response: We replaced the figure 1C and 1D of the original manuscript with new results which showed basal, maximum, and proton leak separately in Figure 1C and 1G of the revised version. In addition, we added ECAR activities of the cell lines for their basal OCR condition (Fig. 1D), media lactate release/glucose consumption (Fig. 1E), and cellular ATP level (Fig. 1F). Revised description for new results was added in lines 91-95, page 4.

Minor comments

2. In figure 2A, please explain how this data was quantified. The graph represents MitoSoxFl %, however, control group Ch-L shows a value of 200%.

Response: It was our mistake. We corrected the graph.

3. Figure 3C, text “NFE2L1” in the second blot overlaps with 130kd in the first blot, please fix it

Response: We fixed the overlapped labels.

4. Most of the methods lacks citation.

Response: We carefully edited overall part of the “Material and Methods,” including citation. The revised parts were highlighted.

5. In methods section 4.3., Western blot analysis, please add information on antibodies dilution, secondary antibodies, and developing methods.

Response: We also edited this part in detail.

6. The statistic is poorly described. No information on tests, N numbers, and significant value is described.

Response: We added description for statistical analysis in “Materials and Methods” (see lines 610-613)

Reviewer 2 Report

This is an interesting paper which reports that a mitochondrial respiratory defect enhances hepatoma invasiveness via STAT3/NFE2L1/STX12 axis.

I have a few comments for the authors:

  1. Although there was evidence of a decreased expression of NDUFA9 did this actually affect the activity of mitochondrial respiratory chain (MRC)  complex I activity?  This is important as it has been reported that only marginal defects in the activities of the MRC complexes can induce oxidative stress.
  2. Can the authors be sure that a deficit in cellular ATP status isn`t involved in NFE2L1 expression? Was cellular ATP status examined?
  3. Was the level of extracellular lactate assessed as this may be involved in the activation of NFE2L1?
  4. Can the authors be certain that this is specific affect of MRC impairment as reactive oxygen species are produced as a byproduct in an number of diseases.
  5. The authors should explain why N-acetlycysteine was used as an antioxidant and not another type of antioxidant.
  6. Can any defect in the MRC induce NFE2L1 expression and if so, why don`t patients with primary mitochondrial disease present with a cancer phenotype?
  7. Would treating cancer cells be counter intuitive  in view of the important of glutathione to cancer cells
  8. The abstract has no actual numerical results and doesn`t explain the relevance of NFE2L1 or how it may be involved in cancer

Author Response

<Response to reviewer 2>

This is an interesting paper which reports that a mitochondrial respiratory defect enhances hepatoma invasiveness via STAT3/NFE2L1/STX12 axis.

I have a few comments for the authors:

Response: We also sincerely appreciate the reviewer for the critical and helpful comments.

1. Although there was evidence of a decreased expression of NDUFA9 did this actually affect the activity of mitochondrial respiratory chain (MRC) complex I activity?  This is important as it has been reported that only marginal defects in the activities of the MRC complexes can induce oxidative stress.

Response: This is a quite important point. So, we compared complex I-dependent OCR by blocking complex II-dependent OCR with 2-thenoyltrifluoroacetone, a complex II inhibitor. We added the result and description in figure 1G of our revised manuscript. (see line 95)

2. Can the authors be sure that a deficit in cellular ATP status isn`t involved in NFE2L1 expression? Was cellular ATP status examined?

Response: When we measured cellular ATP level for the mitochondrial-defective SNU hepatoma cells, SNU354 and SNU423, those cells showed even increased ATP level (Fig. 1F). In addition, these two cell lines showed increased extracellular lactate and glucose consumption (Fig. 1E), indicating that activated glycolytic ATP production compensates the ATP deficit by the mitochondrial defect. We added all these new results in the revised manuscript. (see lines 91-95)

3. Was the level of extracellular lactate assessed as this may be involved in the activation of NFE2L1?

Response: This is another important issue. We also questioned whether increased extracellular lactate s involved in NFE2L1 expression. When we treated 20mM lactate to SNU387 cell, NFE2L1 mRNA was increased as shown the file attached (please see the results in the attached file). However, in the same condition, both intracellular ROS and mitochondrial ROS were increased. These results implied that extracellular lactate may impair OXPHOS and increase ROS. This is another critical issue which requires more detailed study to be fully understood. Therefore, we did not include these results in our revised manuscript.

4. Can the authors be certain that this is specific affect of MRC impairment as reactive oxygen species are produced as a byproduct in a number of diseases.

Response: As mentioned by the reviewer, ROS are produced as a byproduct of diverse cellular activities and those ROS may induce NFE2L1-mediated transcriptional activities as shown in figure 2 where exogenous H2O2 induced NFE2L1 expression. However, final direction of the NFE2L1-mediated cellular activity may be determined differently, depending on cellular context. In the hepatoma cells employed in this study, NFE2L1 did not affect cell growth rate (Fig. 1M and 1N), but regulated cell invasiveness. Here, we tried to emphasize that OXPHOS defect is one of the key ROS generation sources, involved in invasion activity of hepatoma cells with OXPHOS defect.

5. The authors should explain why N-acetlycysteine was used as an antioxidant and not another type of antioxidant.

Response: Although several antioxidants, such as MitoQ and MitoVitE, can remove mitochondrial ROS, we just wanted to prove whether cytoplasmic ROS released from mitochondria can modulate nuclear transcription activity. We added this explanation in the revised manuscript.

6. Can any defect in the MRC induce NFE2L1 expression and if so, why don`t patients with primary mitochondrial disease present with a cancer phenotype?

Response: As described in the first sentence of the results, we previously reported that NFE2L1 mRNA expression is commonly upregulated in response to diverse mitochondrial defects, such as OXPHOS inhibition and mitochondrial DNA depletion (Hepatology, 2015, 62, 1174-1189). At present, the answer to the reviewer’s question is that it is probably due to differential cellular context. As described in the response to the reviewer’s comment #4, either OXPHOS defect or NFE2L1 does not enhance cell proliferation but regulates hepatoma cell invasiveness (Fig. 1M and 1N). In our study, we employed an immortalized liver cell (Ch-L) and hepatoma cells (SNU387, SNU354 and SNU423) which has relatively high cell growth activity. Different cellular context may lead to different pathological phenotypes, such as cancer, neurodegenerative diseases and primary mitochondrial diseases.

7. Would treating cancer cells be counter intuitive in view of the important of glutathione to cancer cells

Response: Although ROS has critical role in cancer development, targeting ROS for anticancer therapy is another big issue. Currently, instead of scavenging cellular ROS, augmenting ROS by pro-oxidant anticancer therapy is importantly regarded as powerful strategy. In addition, multifaceted roles of glutathione have often been reported in both carcinogenesis and anticancer drug resistance. Therefore, it should be carefully approached to treat cancer cells with glutathione in view of therapeutic effectiveness.

8. The abstract has no actual numerical results and doesn`t explain the relevance of NFE2L1 or how it may be involved in cancer.

Response: We improved this point in our revised version (see lines, 32 and 39). We revised the explanation for the relavance of NFE2L1 to cancer that 'NFE2L1 is a key mitochondrial retrograde signaling-mediated primary gene product enhancing hepatoma cell invasiveness via STX12 expression, promoting liver cancer progression' (see lines 42,43).

Round 2

Reviewer 1 Report

All comments are addressed by authors and this manuscript is suitable for publication.